# Synthesis and Evaluation of Peptide–Manganese Dioxide Nanocomposites as Adsorbents for the Removal of Strontium Ions

**DOI:** 10.3390/nano14010052

**Published:** 2023-12-23

**Authors:** Xingjie Lu, Zhen Liu, Wentao Wang, Xin Wang, Hongchao Ma, Meiwen Cao

**Affiliations:** 1State Key Laboratory of Heavy Oil Processing, Department of Biological and Energy Chemical Engineering, College of Chemical Engineering, China University of Petroleum (East China), 66 Changjiang West Road, Qingdao 266580, China; 17864272242@163.com (X.L.); lassiezhen@163.com (Z.L.); z21030169@s.upc.edu.cn (X.W.); 2Department of Radiochemistry, China Institute of Atomic Energy, Beijing 102413, China; wentaowang001@163.com

**Keywords:** peptide assembly, MnO_2_, Sr^2+^ removal

## Abstract

In this study, a novel organic–inorganic hybrid material IIGK@MnO_2_ (2-naphthalenemethyl-isoleucine-isoleucine-glycine-lysine@manganese dioxide) was designed as a novel adsorbent for the removal of strontium ions (Sr^2+^). The morphology and structure of IIGK@MnO_2_ were characterized using TEM, AFM, XRD, and XPS. The results indicate that the large specific surface area and abundant negative surface charges of IIGK@MnO_2_ make its surface rich in active adsorption sites for Sr^2+^ adsorption. As expected, IIGK@MnO_2_ exhibited excellent adsorbing performance for Sr^2+^. According to the adsorption results, the interaction between Sr^2+^ and IIGK@MnO_2_ can be fitted with the Langmuir isotherm and pseudo-second-order equation. Moreover, leaching and desorption experiments were conducted to assess the recycling capacity, demonstrating significant reusability of IIGK@MnO_2_.

## 1. Introduction

Nuclear energy is widely used due to its ability to reduce greenhouse effects, high energy density, and ease of storing nuclear fuel [1]. However, in nuclear disasters such as Chernobyl and Fukushima, the release of large amounts of radioactive isotopes and waste has had a catastrophic impact on ecosystems, causing significant damage. In terms of radioactive safety, strontium (^90^Sr), cobalt (^60^Co), and cesium (^137^Cs) are considered the main radioactive isotopes due to their relatively long half-lives, high solubility, and easy transfer [2]. Among them, ^90^Sr is a carcinogen and hazardous pollutant that has chemical properties similar to calcium and is easily absorbed by the human body [3]. Therefore, the development of new materials for removing strontium ions (Sr^2+^) from aqueous solutions has received great attention. At present, various methods have been developed to remove radioactive toxic ions from aqueous solutions, including chemical precipitation, membrane, solvent extraction, ion exchange, and adsorption [4,5,6,7,8]. Among them, the adsorption method is the most used and highly efficient technology. Up to now, a large number of nanomaterials, such as hydroxyapatite nanoparticles, metal sulfide, metallic oxide, graphene oxide (GO), and MOFs, have been employed as adsorbents for removing Sr^2+^ [9,10,11,12,13]. Attributed to their larger specific surface area and more active sites, MnO_2_ nanoparticles have been widely studied and applied as effective adsorbents for Sr^2+^ [14]. Moreover, MnO_2_ particles also exhibit selectivity for Sr^2+^ in the presence of competing ions, including Na^+^, K^+^, Ca^2+^, and Mg^2+^ [15]. However, MnO_2_ nanoparticles have drawbacks such as susceptibility to van der Waals forces, which may lead to aggregation and a decrease in their adsorption performance [16]. Therefore, it is necessary to fix MnO_2_ nanoparticles on the carrier to improve their dispersion stability as well as adsorption performance and reusability for Sr^2+^ removal.

Due to the presence of abundant functional groups such as hydroxyl groups, thiol groups, and carboxyl groups as adsorption sites, along with diverse secondary structures, a series of peptides has been studied for their binding to various metallic materials, including metal oxides, metal sulfides, and metals [17,18]. Moreover, some specifically designed peptides demonstrate a selective affinity for particular metal ions, which can serve as effective metal-binding moieties in aqueous solutions and promote selective metal ion adsorption [19]. For example, Mondal et al. confirmed the affinity of a tripeptide gel for metal ions, including lead and cadmium [20]. Besides direct metal ion adsorption, the peptides can be used to composite with other materials to achieve synergistic metal ion adsorption. At present, various peptide sequences have been used for biomimetic synthesis of ceramics, metal sulfides, metal oxides, and metal nanoparticles, and composite materials with low dispersion, good crystallinity, diverse morphology, and excellent biological functions have been constructed in mild water environments [21,22]. In addition, peptides have excellent assembly performance and can form different functional structures through covalent and noncovalent interactions (such as hydrophilicity and hydrophobicity, π-π stacking, hydrogen bonding, and electrostatic interactions). These peptide assembly structures can regulate the structure of peptide–inorganic nanoparticles by inducing the nucleation and growth process of inorganic nanoparticles and selectively binding to certain crystal planes. This makes peptides an excellent template for in situ loading and construction of inorganic nanomaterials.

As mentioned in the above text, though MnO_2_ nanoparticles are good candidates for Sr^2+^ absorption and removal, they easily experience severe agglomeration and poor dispersibility in aqueous solution, which influences the adsorption performance. To address this issue, we chose the self-assembled nanofibrils of short peptide 2-naphthalenemethyl-isoleucine-isoleucine-glycine-lysine (IIGK) (Figure 1) as templates to mediate the mineralization of MnO_2_ on their surface, which can greatly enhance the dispersibility of MnO_2_ nanoparticles. Compared with other methods for synthesizing MnO_2_, such as electrodeposition, demulsification, and hydrothermal methods, this method is environmentally friendly, mild, and easy to operate. The physicochemical properties of the as-prepared IIGK@MnO_2_ were systematically characterized first by using various techniques. Then, experiments were performed to investigate the adsorption performance of the IIGK@MnO_2_ nanocomposites towards Sr^2+^. The results show that the IIGK@MnO_2_ nanocomposites have a strong affinity and adsorption capacity for Sr^2+^, and a synergistic effect on the adsorption of Sr^2+^ with IIGK@MnO_2_ can be concluded. This study demonstrated the merits of IIGK@MnO_2_ as a promising adsorbent to remove Sr^2+^ from nuclear wastewater.

## 2. Materials and Methods

### 2.1. Materials

Manganese chloride tetrahydrate (MnCl_2_·4H_2_O), potassium permanganate (KMnO_4_), sodium nitrate (NaNO_3_), potassium nitrate (KNO_3_), strontium nitrate (Sr(NO_3_)_2_), sodium hydroxide (NaOH), and potassium hydroxide (KNO_3_) were acquired from Sinopharm Chemical Reagent Co., Ltd. (Shanghai, China). All working solutions were prepared using distilled water. The pH was adjusted using NaOH and HCl as needed and monitored with a digital pH meter. All adsorption experiments were conducted at ambient temperature.

### 2.2. Preparation of IIGK@MnO_2_ Nanocomposite

First, 300 mg IIGK powder was dissolved in 250 mL ultrapure water, which was then subjected to ultrasonic dispersion for approximately 10 min. Then, the mixture was left to stand overnight, followed by adding another 225.6 mL ultrapure water. Subsequently, 9.6 mL KmnO_4_ (30 mM) aqueous solution and 9.6 mL MnCl_2_·4H_2_O (42 mM) aqueous solution were added to the mixture, which was then stirred for 24 h. Finally, the samples were centrifuged, washed three times with ultrapure water, and subsequently freeze-dried to obtain IIGK@MnO_2_ powder for further use.

### 2.3. Characterization

The morphology of samples was characterized using a scanning electron microscope (SEM, Hitachi S-4800, Tokyo, Japan) and transmission electron microscope (TEM, JOEL JEM1400 Plus, Tokyo, Japan). The SEM equipped with an Oxford Instruments energy-dispersive X-ray spectroscopy (EDS) system was used for analyzing elemental composition. The thickness of materials was measured using atomic force microscopy (AFM, Santa Babara Vecco Nanoscope Iva, Santa Barbara, CA, USA). The size and zeta potential of samples were determined on a Zetasizer Nano (DLS, Malvern NANO ZS, Marvin City, UK). The element analysis of the samples before and after Sr^2+^ adsorption was carried out using X-ray photoelectron spectroscopy (XPS, Thermo Scientific Esclab 250Xi, Waltham, MA, USA) with a monochromatic Al Kα X-ray source (15 KV). Fourier-transform infrared spectroscopy (FT-IR, Thermo Scientific Nicolet iS5, Waltham, MA, USA) was performed to investigate the interaction between MnO_2_ and IIGK. X-ray diffraction (XRD, PANalitical X‘Pert PRO MPD, Almelo, The Netherland) was performed to study the crystal structure of the samples using Cu-Kα radiation in the 2θ range of 10°–80° at room temperature. N_2_ adsorption–desorption isotherm was conducted on the gas sorption analyzer at 373.15 K (Malvern PANalitical Autosorb-6B, Marvin City, UK), and the specific area and the average pore diameters were stimulated using the Brunauer-Emmett-Teller (BET) and Barrett–Joyner–Halenda (BJH) method. Inductively coupled plasma optical emission spectroscopy (ICP-OES, HORIBA JY 2000-2, Loos, Naples, FL, USA) was employed to determine the concentration of residual metal ions in the solution.

### 2.4. Adsorption Performance of IIGK@MnO_2_

All adsorption experiments were carried out in 15 mL polyethylene centrifuge tubes. After mixing 1 mg IIGK@MnO_2_ with 10 mL Sr^2+^ aqueous solution, the pH of the mixture was adjusted to a suitable value. The mixture was left in a constant-temperature shaker for a certain time, followed by centrifugation at 12,000 rpm for 20 min to extract the supernatant. Finally, the residue Sr^2+^ concentration in the adsorbed solution was measured using ICP-OES. The equilibrium adsorption amount (*q*_e_, mg/g) and removal rate (*R*, %) of IIGK@MnO_2_ towards Sr^2+^ can be calculated by the following equations:(1)$$q_{e}=(c_{0}-c_{e})\times \frac{V}{M}$$
(2)$$R=\frac{c_{0}-c_{e}}{c_{0}}\times 100\%$$
where *c*_0_ (mg/L) and *c*_e_ (mg/L) are the initial and equilibrium concentrations of Sr^2+^ in the solution, respectively. *V* (mL) represents the volume of the adsorbed solution, and *M* (g) is the quantity of adsorbent.

### 2.5. Desorption of Strontium Ions from IIGK@MnO_2_

The desorption behavior of the IIGK@MnO_2_ was investigated by treating IIGK-MnO_2_-Sr composites with 1 M HCl aqueous solution. The amount of Sr^2+^ desorbed from the IIGK@MnO_2_, *Q*_D_Sr_, was determined according to Equation (3), while the desorption efficiency (*D*e, %) was obtained from Equation (4).
(3)$$Q_{D\_Sr}=\frac{C_{D}\times V_{D}}{M_{D}}$$
(4)$$De=\frac{Q_{D\_Sr}}{q_{e}}\times 100$$
where *C*_D_ (mg/L) is the equilibrium desorption concentrations of Sr^2+^, *V*_D_ (mL) represents the volume of the eluent, and *M*_D_ (g) is the weight of adsorbent after Sr^2+^ adsorption.

## 3. Results and Discussion

### 3.1. Synthesis and Characterization of IIGK@MnO_2_ Nanocomposite

In this work, the IIGK@MnO_2_ nanocomposite was synthesized by depositing MnO_2_ nanoparticles along peptide fibers using a green, simple, and easy-to-operate biomimetic mineralization method. For comparison, pure manganese dioxide nanoparticles without peptide involvement were also synthesized. The morphology of MnO_2_ nanoparticles and IIGK@MnO_2_ nanocomposites was characterized using SEM, TEM, and AFM. As shown in Figure 2a,b, the MnO_2_ nanoparticles generated without peptide show a disordered flower shape with a diameter of 400–500 nm, while the IIGK@MnO_2_ nanocomposites took on elongated structures with many MnO_2_ nanoparticles aligned in an axial direction (Figure 2c,d). The TEM result (Figure 2e) shows more clearly the one-dimensional nanowire/nanosheet composited structure. The results indicate that MnO_2_ is generated through biomimetic mineralization using IIGK short peptides as templates. The inner core should be the IIGK fibrils, while the outer layer sheets should be MnO_2_ nanostructures. The results demonstrate clearly the template role of the peptide fibrils in mediating MnO_2_ mineralization. The formation of fibrous IIGK@MnO_2_ nanocomposite can be confirmed with an AFM image (Figure 2f).

The surface zeta potential of the one-dimensional fiber-like IIGK@MnO_2_ nanocomposites at different pH was then measured, as shown in Figure 3a. It can be observed that the isoelectric point of IIGK@MnO_2_ appears at pH 1.7. While the pH value increases from 0.9 to 10.5, the zeta potential of IIGK@MnO_2_ decreases gradually from 10 mV to −40 mV. Notably, when the pH value is greater than 1.7, IIGK@MnO_2_ undergoes charge reversal, changing from a positively charged surface to a negatively charged surface. Moreover, the excess negative charges on the IIGK@MnO_2_ surface are beneficial for the subsequent adsorption of Sr^2+^. The particle size distribution of MnO_2_ and IIGK@MnO_2_ was measured using the dynamic light scattering measurement. As shown in Figure 3b, the size of IIGK-induced IIGK@MnO_2_ nanocomposites is about 100–400 nm, while that of MnO_2_ produced in the absence of peptide is above 1000 nm. This result indicates that the IIGK@MnO_2_ nanocomposites induced with IIGK have a higher dispersion stability in the solution and a larger specific surface area, which is beneficial for the adsorption of Sr^2+^.

In order to investigate the crystal structure, X-ray powder diffraction (XRD) was employed to characterize IIGK@MnO_2_ and MnO_2_. The XRD patterns of the IIGK@MnO_2_ nanocomposites and MnO_2_ itself are shown in Figure 3c. For MnO_2_, the peaks are found at 2θ = 22.8°, 36.5°, and 65.7°. These crystalline microregions correspond to δ-standard diffraction patterns (002), (111), and (311) of MnO_2_ (JCPDS 80-1098), respectively. The same peaks were found in the XRD patterns of IIGK@MnO_2_, indicating that MnO_2_ was successfully loaded on the IIGK fibril’s surface. Moreover, as indicated by the weak XRD peaks, the IIGK@MnO_2_ nanocomposites show a low crystallinity level. The reason for the poor crystallinity may be due to the fast reaction rate during the interaction between KMnO_4_ and MnCl_2_, which leads to the rapid formation and precipitation of MnO_2_ [23]. This rapid process may have hindered the formation of an orderly atomic arrangement.

Subsequently, the surface area, pore size, as well as pore volume of the IIGK@MnO_2_ and MnO_2_ were determined using BET analysis and the BJH method. According to the adsorption–desorption isotherms shown in Figure 3d, the surface area of IIGK@MnO_2_ is approximately 46.19 m^2^·g^−1^. In addition, the pore volume and average pore size of IIGK@MnO_2_ are calculated to be 2.40 cm^3^·g^−1^ and 14.02 nm, respectively. The whole results also indicate that the IIGK@MnO_2_ nanocomposite has suitable physical properties as an adsorbent and is suitable for the adsorption of Sr^2+^ [24].

Furthermore, Fourier transform infrared spectroscopy (FTIR) was used to characterize IIGK@MnO_2_ at a molecular level (Appendix A). The broad band at about 3421.6 cm^−1^ belongs to the stretching vibration of -OH, which may be attributed to the adsorption of a small amount of water by IIGK@MnO_2_ and the presence of hydroxyl groups on the Mn surface [25]. In addition, the characteristic absorption peaks at 1628.2 cm^−1^, 1383.7 cm^−1^, and 1022.8 cm^−1^ can be attributed to the bending vibration of the hydroxyl group (Mn-OH) directly connected to the Mn atom. The peak appearing at 500 cm^−1^ indicates the presence of [MnO_6_] octahedra, which is caused by the stretching of Mn-O and Mn-O-Mn bonds in the octahedral structure [26]. The elements’ valence states of IIGK@MnO_2_ were further characterized using XPS spectroscopy. As shown in Figure 4a, the XPS spectrum of Mn 2p exhibits two peaks at 642.7 eV and 654.4 eV, corresponding to Mn 2p 3/2 and Mn 2p 1/2, respectively. The spin energy range of 11.7 eV indicates that the IIGK short peptide fibers are mainly used as templates for the deposition of MnO_2_ [27]. In addition, the weak peak appearing at 645.6 eV indicates the presence of a small amount of unreacted Mn^2+^ in the system. According to Chigane et al., the XPS spectrum of Mn 3s can split into two weaker peaks, which are located at 84.6 eV and 89.5 eV, corresponding to Mn^4+^ and Mn^3+^, respectively [28]. The difference in binding energy between these two peaks increases with the decrease in the valence state of Mn. As shown in Figure 4b, a 4.9 eV difference can be observed between the two peaks, indicating that Mn in IIGK@MnO_2_ is mainly composed of Mn^4+^, with a small amount of Mn^3+^ present. The O 1s peak exhibits three resolved peaks, with binding energies concentrated at 530.1 eV, 531.4 eV, and 533.12 eV (Figure 4c). These peaks are attributed to Mn-O-Mn, Mn-O, and O-H, respectively [29]. Figure 4d shows the complete XPS spectrum of IIGK@MnO_2_, including Mn 3s, Mn 2p, C 1s, O 1s, and N 1s signals, indicating the successful mineralization of MnO_2_.

### 3.2. Sr^2+^ Adsorption Studies

Previous reports have indicated that MnO_2_ possesses excellent adsorbing performance for Sr^2+^. Then, the performance of IIGK@MnO_2_ in adsorbing Sr^2+^ was studied. The ability of IIGK@MnO_2_ to adsorb Sr^2+^ at different pH values was first evaluated. As shown in Figure 5a, the adsorption efficiency of Sr^2+^ increases with the increase in pH value and then tends to equilibrium. The low adsorption efficiency of IIGK@MnO_2_ for Sr^2+^ in acidic pH environments may be caused by the competitive interaction between H^+^ and Sr^2+^. At higher pH conditions, the concentration of H^+^ is lower; thus, there are more active sites left on IIGK@MnO_2_ for the adsorption of Sr^2+^. In addition, under high pH conditions, the surface of IIGK@MnO_2_ carries more negative charges, which has a stronger electrostatic force on positively charged Sr^2+^, thereby improving the adsorption efficiency of Sr^2+^ [30]. Except for pH, temperature also affects the adsorption efficiency of IIGK@MnO_2_ for Sr^2+^. As shown in Figure 5b, IIGK@MnO_2_ has the highest adsorption efficiency for Sr^2+^ at room temperature. When the temperature rises above 40 °C, the adsorption efficiency sharply decreases. This phenomenon may be due to the weakening of the interaction between IIGK@MnO_2_ and Sr^2+^ at higher temperatures, resulting in the desorption of Sr^2+^ [31].

The contact time between IIGK@MnO_2_ and Sr^2+^ would also affect the adsorption efficiency. As shown in Figure 5c, the adsorption of Sr^2+^ by IIGK@MnO_2_ rapidly reaches equilibrium at 6 h. The initial rapid adsorption may be attributed to the large number of available adsorption sites exposed on the surface of IIGK@MnO_2_. Then, the adsorption process reached equilibrium after 6 h, indicating that most of the available adsorption sites were occupied. The whole adsorption process was assessed using both the pseudo-first-order and pseudo-second-order kinetic models. The equations are as follows [32]:(5)$$lg\left(q_{e}-q_{t}\right)\;=lgq_{e}-\frac{k_{1}}{2.303}$$
(6)$$\frac{t}{q_{t}}=\frac{1}{k_{2}q_{e}^{2}}+\frac{1}{q_{e}}$$
where *q*_e_ and *q*_t_ represent the Sr^2+^ adsorption capacity (mg/g) at equilibrium or at time *t* (adsorption time), respectively; *k*_1_ (min^−1^) and *k*_2_ (g/(mg∙min)) represent the rate constants for the pseudo-first-order and pseudo-second-order models. The fitted parameters are summarized in Table 1. Comparing the R^2^ value of these models, the pseudo-second-order model fits better than the pseudo-first-order model [33].

For adsorbents, the initial concentration of the adsorbed substance often affects the adsorption rate and efficiency. Therefore, the effect of initial Sr^2+^ concentration on the adsorption efficiency of IIGK@MnO_2_ was further studied. According to Figure 5d, the adsorption efficiency of IIGK@MnO_2_ for Sr^2+^ shows a trend of first increasing and then decreasing with the increase in initial Sr^2+^ concentration. The maximum adsorption efficiency occurs when the Sr^2+^ concentration is 25 mg·L^−1^. When the initial Sr^2+^ concentration reaches 40 mg·L^−1^, the adsorption efficiency sharply decreases. When the Sr^2+^ concentration increases to 60 mg·L^−1^, the adsorption efficiency tends to stabilize. The adsorption isotherms of Sr^2+^ on IIGK@MnO_2_ were studied using Freundlich and Langmuir isotherm models. The model is shown as follows [34]:(7)$$q_{e}=\frac{q_{m}bC_{e}}{1+bC_{e}}$$
(8)$$q_{e}={\;K}_{f}C_{e}^{\frac{1}{n}}$$
where *q*_m_ (mg/g) represents the maximum adsorption capacity at the isotherm temperature, *b* (L/mg) is the Langmuir constant associated with the free energy or net enthalpy of adsorption, *C*_e_ (mg/L) and *q*_e_ (mg/g) denote the concentration of the adsorbate in the equilibrated solution and the amount adsorbed on the adsorbent, respectively, *K*_f_ and *n* are equilibrium constants that provide information about the adsorption capacity and intensity. Shown in Table 2 is the fitting effect of equilibrium data, and the Langmuir isotherm model (R^2^ = 0.991) is better than that of the Freundlich model (R^2^ = 0.982). Furthermore, the Sr^2+^ adsorption performance of the IIGK@MnO_2_ nanocomposite was compared with that of other reported materials (Table 3). Compared to other adsorbents, IIGK@MnO_2_ showed better performance in the aspects of higher adsorption capacity (748.2 mg/g), which is beneficial for the treatment of nuclear wastewater. Therefore, the adsorbent surface is considered homogeneous and monolayer in terms of adsorption energy [35]. Compared with pure MnO_2_, IIGK@MnO_2_ composites exhibit significantly stronger adsorption efficiency for Sr^2+^ (Figure 5e). As previously speculated, after combing IIGK peptide with MnO_2_, the obtained IIGK@MnO_2_ composites contain more adsorption active sites due to their high dispersion, large specific surface area, and high negative charge, which is beneficial for the adsorption of Sr^2+^.

### 3.3. Desorption and Reusability Studies

When evaluating the economy and applicability of an adsorbent, the reusability is an important indicator that plays a crucial role in the practical application of adsorbents. Herein, the adsorption/desorption process was repeated to assess the reusability of IIGK@MnO_2_. Firstly, IIGK@MnO_2_ was used to adsorb Sr^2+^, and then the IGK@MnO_2_-Sr^2+^ composite was treated with 1M HCl solution for Sr^2+^ desorption. The whole adsorption/desorption process was repeated three times. As shown in Figure 6, when the regeneration cycle is repeated three times, the adsorption efficiency of IIGK@MnO_2_ for Sr^2+^ is still higher than 64%, indicating that 1M HCl solution can effectively desorb Sr^2+^ from IIGK@MnO_2_. Moreover, the adsorption capacity of IIGK@MnO_2_ for Sr^2+^ decreased with the increase in cycling number, which may be due to H^+^ occupying some adsorption sites. However, it can also be noted that the adsorption efficiency of IIGK@MnO_2_ for Sr^2+^ can still reach over 60% at the end of three cycles, indicating that IIGK@MnO_2_ has good reusability.

### 3.4. Adsorption Mechanism of Sr^2+^ by the IIGK@MnO_2_ Nanocomposite

The adsorption mechanism of IIGK@MnO_2_ for Sr^2+^ was studied. Figure 7a shows the morphology and distribution of various elements of IIGK@MnO_2_ after the adsorption of Sr^2+^. After adsorbing Sr^2+^, there was no significant change in the morphology of IIGK@MnO_2_. The mapping of IIGK@MnO_2_-Sr elements shows that the adsorbed Sr^2+^ is evenly distributed on the surface of IIGK@MnO_2_, indicating that the adsorption sites on IIGK@MnO_2_ are relatively uniform. XPS data confirmed the presence of strontium on IIGK@MnO_2_ after adsorption (Figure 7b). The deconvolution peaks located at 133.5 eV and 135.4 eV belong to Sr 3d 5/2 and Sr 3d 3/2, respectively (Figure 7c) [10]. There is a shift in binding energy in the high-resolution spectrum of O1s in Figure 7d. The binding energy of the O1s-Mn-OH peak decreased from 531.4 eV to 531.0 eV, and the binding energy of the O 1s O-H peak decreased from 533.1 eV to 532.8 eV. This result indicates that oxygen atoms interact with Sr^2+^ and play a dominant role in the Sr^2+^ adsorption process.

From the above results, the mechanism of Sr^2+^ adsorption by IIGK@MnO_2_ can be proposed, as schemed in Figure 7e. On the one hand, the peptide IIGK can bind with Sr^2+^ through coordination or electrostatic interactions. On the other hand, the MnO_2_ nanoparticles on the IIGK fibrils’ surface have many absorption sites, which can adsorb Sr^2+^ by either coordination interaction or ion exchange reaction. Therefore, IIGK and MnO_2_ work synergistically to enhance the removal efficiency of the IIGK@MnO_2_ nanocomposites.

## 4. Conclusions

In this work, the IIGK@MnO_2_ nanocomposites were synthesized using environmentally friendly and mild methods. Using the short peptide IIGK as a template leads to the formation of a one-dimensional fibrous structure of IIGK@MnO_2_ nanocomposite, which not only gives the mineralized IIGK@MnO_2_ complex a large specific surface area but also makes the overall IIGK@MnO_2_ negatively charged. These unique properties are conducive to the adsorption and removal of Sr^2+^ from wastewater. According to the adsorption performance of IIGK@MnO_2_ towards Sr^2+^, it can be observed that the IIGK@MnO_2_ hybrid adsorbents possess excellent Sr^2+^ adsorption effects in a large range of temperature and pH values and show much higher Sr^2+^ adsorption efficiency than pure MnO_2_ nanoparticles. In addition, IIGK@MnO_2_ also exhibits good reusability, achieving a Sr^2+^ adsorption efficiency of over 60% after three cycles of adsorption–desorption processes. By analyzing the element distribution and valence states of IIGK@MnO_2_-Sr after the adsorbing process, the mechanism of Sr^2+^ adsorption was proposed. The interaction between N-H groups on IIGK and oxygen atoms on MnO_2_ with Sr^2+^ may play an important role in the adsorption of strontium ions. In conclusion, the designed IIGK@MnO_2_ nanocomposite in this work exhibits excellent adsorption performance towards Sr^2+^ and will shed light on the construction of organic–inorganic hybrid adsorbents with multiple active adsorption sites and high adsorption efficiency for adsorbing radioactive ions in wastewater.

## Figures and Tables

**Figure 1 nanomaterials-14-00052-f001:**
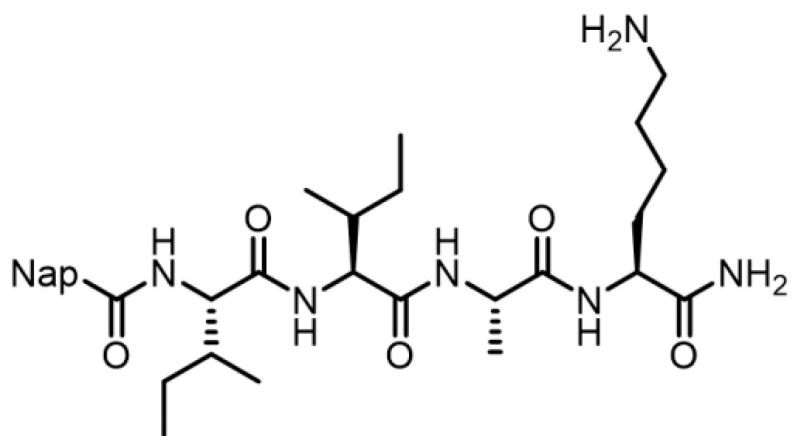
Molecular structure of the short peptide IIGK.

**Figure 2 nanomaterials-14-00052-f002:**
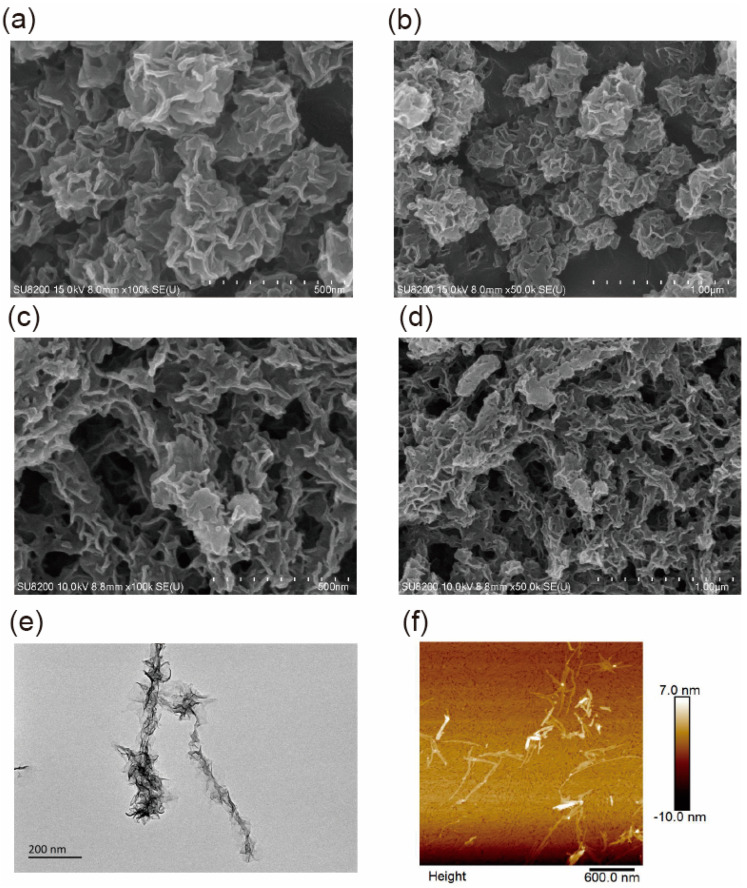
(**a**,**b**) SEM images of MnO_2_, (**c**,**d**) SEM images of IIGK@MnO_2_, (**e**) TEM, and (**f**) AFM images of IIGK@MnO_2_ nanocomposite.

**Figure 3 nanomaterials-14-00052-f003:**
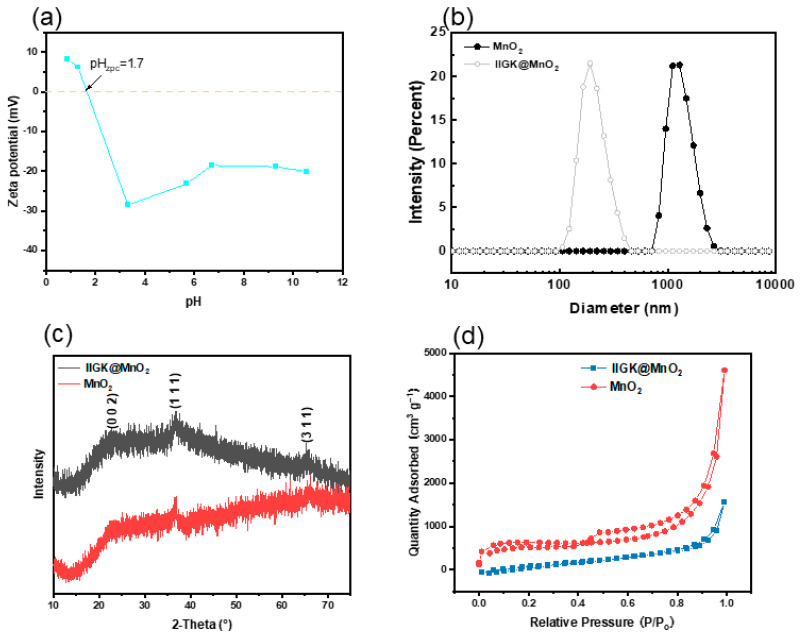
(**a**) Zeta potential of IIGK@MnO_2_, (**b**) diameter distribution of MnO_2_ and IIGK@MnO_2_, (**c**) XRD spectra of IIGK@MnO_2_ and MnO_2_, and (**d**) N_2_ adsorption–desorption isotherms of IIGK@MnO_2_ and MnO_2_.

**Figure 4 nanomaterials-14-00052-f004:**
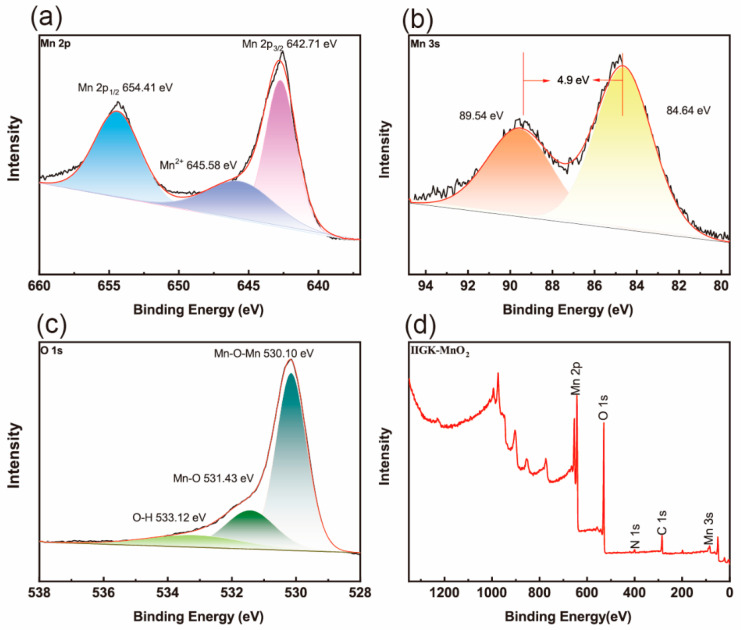
XPS spectra of (**a**) Mn 2p, (**b**) Mn 3s, and (**c**) O 1s of IIGK@MnO_2_. (**d**) XPS full survey spectra of IIGK@MnO_2_. The fitting data of spectra (**a**–**c**) are shown by different colors.

**Figure 5 nanomaterials-14-00052-f005:**
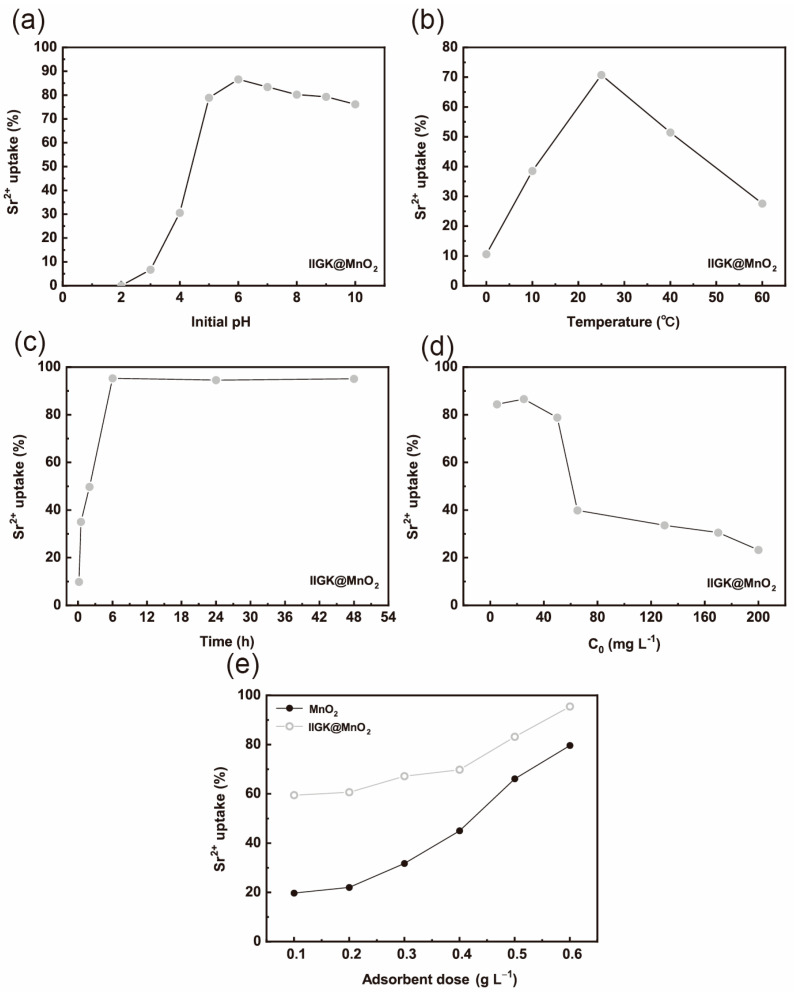
Sr^2+^ removal efficiency of the IIGK@MnO_2_ nanocomposite at different (**a**) pH, (**b**) temperature, (**c**) contact time, (**d**) initial Sr^2+^ concentration, and (**e**) different adsorbent dosage.

**Figure 6 nanomaterials-14-00052-f006:**
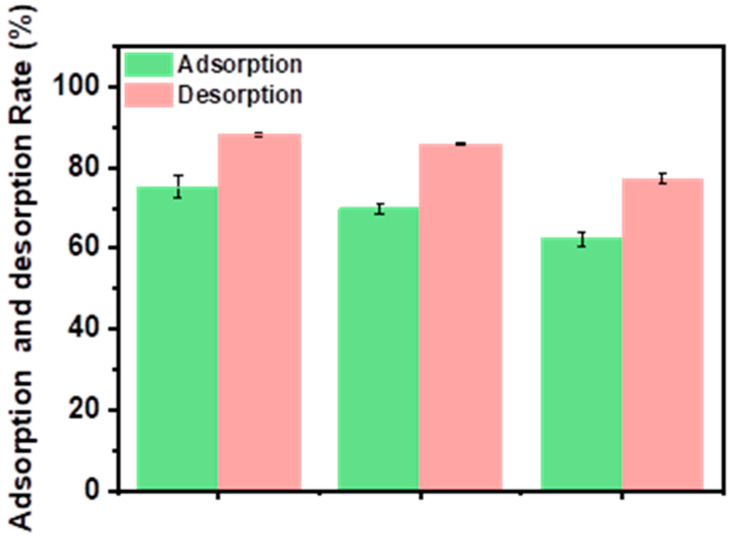
Reusability of the IIGK@MnO_2_ nanocomposite during three cycles of Sr^2+^ adsorption and desorption.

**Figure 7 nanomaterials-14-00052-f007:**
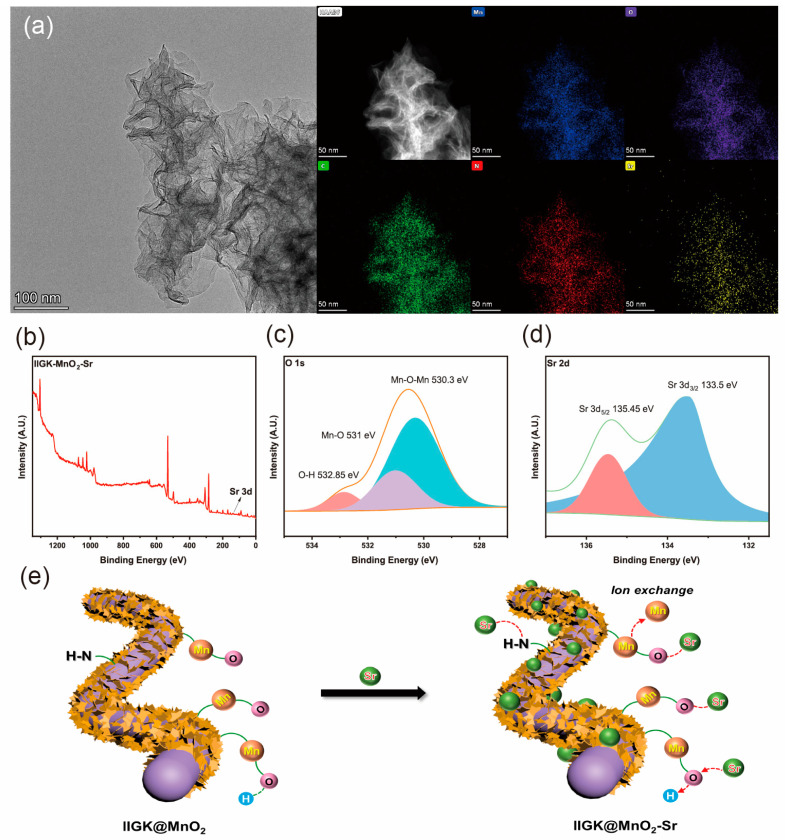
(**a**) TEM imagine and EDS element-mapping of IIGK@MnO_2_-Sr composite. (**b**) XPS full survey spectra of the IIGK@MnO_2_-Sr nanocomposite. (**c**,**d**) XPS fine spectra of O 1s and Sr 3d of the IIGK@MnO_2_-Sr nanocomposite. (**e**) Schematic diagram of the adsorption mechanism of Sr^2+^ by IIGK@MnO_2_.

**Table 1 nanomaterials-14-00052-t001:** Fitted parameters of pseudo-first-order and pseudo-second-order kinetic models.

Model	Parameters	Value
Pseudo-First-Order	*q*_e_ (mg/g)	238.94
	*k*_1_ (min^−1^)	0.489
	R^2^	0.939
Pseudo-Second-Order	*q*_e_ (mg/g)	252.542
	*k*_2_ (g/mg·min)	0.003
	R^2^	0.936

**Table 2 nanomaterials-14-00052-t002:** The parameters calculated from the Langmuir and Freundlich for Sr^2+^ adsorption by the IIGK@MnO_2_ nanocomposite.

Models	Parameters		
Langmuir isotherm	*q*_m_ (mg/g)	b (L/mg)	R^2^
	748.193	0.015	0.991
Freundlich isotherm	k	1/n	R^2^
	33.542	0.55	0.982

**Table 3 nanomaterials-14-00052-t003:** Comparison of the adsorption performance of IIGK@MnO_2_ nanocomposite and other adsorbents for removal of Sr^2+^.

Adsorbent	Maximum Adsorption Capacity (mg/g)	Optimal pH	Equilibrium Time (min)	Reference
IIGK@MnO_2_	748.193	6	360	This study
bentonite	63.01	-	-	[36]
zeolite 4A	252.5	4–9	5	[37]
K_2_SbPO_6_	175.9	3	1440	[38]
SBA-15	17.67	10	100	[39]
A-ZrP	300	3–11	240	[40]
NaTS	80.0	5	0.083	[10]
SZ-6	61.4	10	50	[41]
MnO_2_	53.5	10	0.25	[24]

## Data Availability

All data generated or analyzed during this study are contained within the article and Appendix A.

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
