# Peer review of "Synthesis and Evaluation of Peptide–Manganese Dioxide Nanocomposites as Adsorbents for the Removal of Strontium Ions"

_nanomaterials, 2023, doi:10.3390/nano14010052_

Round 1

Reviewer 1 Report

Comments and Suggestions for Authors

Manuscript No: nanomaterials-2776978

Title: Synthesis and evaluation of peptide-manganese dioxide nano-composites as adsorbents for the removal of strontium ions

The following criticisms should be taken into consideration by the author to enhance the work. 

1.      The introduction should also include the full form of IIGK.

2.      It would be better if the author provided a literature review on the peptide-based adsorbent utilized for the removal of pollutants or strontium. Then, elaborate on why peptide is a superior adsorbent for strontium elimination when modified MnO2?

3.      In the introduction, the author should discuss the uniqueness and novelty of the present study.

4.      4. Using TEM and AFM to determine the morphology of composite materials is highly challenging. Therefore, the author should provide SEM images of the IIGK@MnO2 nanocomposite at a resolution of 500 nanometers to 1 micrometer.

5.      In general, zeta potential or point of zero charge measurements are conducted at a range of pH values. It is pointless to discuss the zeta potential of adsorbents if one is unsure of the charges at a specific pH. What was the pH at which the adsorbents acquired negative charges? It needs attention.

6.      It would be preferable if the author provided discussion on MnO2's XRD and BET.

7.      Since the amount of gas adsorption is more than 1500 cm3/g, I'm excited about seeing the BET result of IIGK@MnO2. So why does this adsorbent have such a small surface area (46.1 m2/g)? Type 1 isotherm (microporous materials)-based adsorbents may typically achieve a gas adsorption volume of more than 1500 cm3/g, and the surface area may also be able to exceed 1000 m2/g.

8.      What is this “there are some crystalline micro areas on the surface”? The author needs to give it more consideration.

9.      It would be preferable if the author discussed the adsorption experiments to the MnO2, including variables such as pH, temperature, time, and dye concentration.

10.   At least five cycles of reusability testing should be performed for adsorption studies.

11.   What caused the author to select HCl with a high molarity for reusability purposes? Regenerated agents that are low in toxicity or non-toxic are available or can be made. Acids are generally not advised for use in reusability studies due to their highly toxic and harmful impacts on both the environment and human health.

12.   I agree with the author that TEM and XPS investigations were utilized to clarify the adsorption mechanism. However, it is also necessary to include a schematic representation of potential interactions between the adsorbent and pollutant structures when discussing the adsorption mechanism. Then it looks according to scientific standards and is understandable to the reader who works in the field.

13.   The conclusion lacks coherence and relevance. It should also include a discussion of the primary results. In addition to the challenges, future implantation may be suggested.

Author Response

Thank the reviewer greatly for the valuable comments. The point-by-point response can be found in the attached file.

Reviewer 2 Report

Comments and Suggestions for Authors

The authors report the preparation of MnO2 supported by a short peptide chain. The authors provide ample characterization for the material and follow with an adsorption study of strontium ions. The results are presented clearly but I have a few minor issues that should be addressed prior to publication.  

Table 3 illustrates a problem in this area. The authors summarize a family of potential adsorbents, but the conditions between all studies a significantly different. It would be ideal if the authors attempted to replicate one in an effort to support some kind of benchmarking in this area. I think the authors can make that choice. For example, they could replicate the use of just MnO2 under their optimal conditions, but any adsorption study from bulk water should be presumably conducted near pH 7.

It is also unclear what factor or factors impact this adsorption behavior versus other materials. It is clear from Figure 6 that there is an inherent dependence on a gross conformer of the polypeptide. Does IIGK have substantial absorption, for example?    

The infrared spectrum (Figure 4) could be presented in the supporting information, and the space recovered would be better used to make Figure 6 larger and more readable.

Author Response

Thank the reviewer very much for the valuable comments. The point-by-point response can be found in the attached file.

Round 2

Reviewer 1 Report

Comments and Suggestions for Authors

Accept